# Two Novel, Flavin-Dependent Halogenases from the Bacterial Consortia of *Botryococcus braunii* Catalyze Mono- and Dibromination



**Pia R. Neubauer** [1], **Olga Blifernez-Klassen** [2], **Lara Pfaff** [1], **Mohamed Ismail** [1], **Olaf Kruse** [2] **and Norbert Sewald** [1,*]

[1] Organic and Bioorganic Chemistry, Faculty of Chemistry and Centre for Biotechnology (CeBiTec), Bielefeld University, D-33615 Bielefeld, Germany; pia.neubauer@uni-bielefeld.de (P.R.N.); lara.pfaff@uni-greifswald.de (L.P.); M.Ismail10@bradford.ac.uk (M.I.)

[2] Algae Biotechnology and Bioenergy, Faculty of Biology and Centre for Biotechnology (CeBiTec), Bielefeld University, D-33615 Bielefeld, Germany; olga.blifernez@uni-bielefeld.de (O.B.-K.); olaf.kruse@uni-bielefeld.de (O.K.)

\* Correspondence: norbert.sewald@uni-bielefeld.de; Tel.: +49-521-106-6963

**Abstract:** Halogen substituents often lead to a profound effect on the biological activity of organic compounds. Flavin-dependent halogenases offer the possibility of regioselective halogenation at non-activated carbon atoms, while employing only halide salts and molecular oxygen. However, low enzyme activity, instability, and narrow substrate scope compromise the use of enzymatic halogenation as an economical and environmentally friendly process. To overcome these drawbacks, it is of tremendous interest to identify novel halogenases with high enzymatic activity and novel substrate scopes. Previously, Neubauer et al. developed a new hidden Markov model (pHMM) based on the PFAM tryptophan halogenase model, and identified 254 complete and partial putative flavin-dependent halogenase genes in eleven metagenomic data sets. In the present study, the pHMM was used to screen the bacterial associates of the *Botryococcus braunii* consortia (PRJEB21978), leading to the identification of several putative, flavin-dependent halogenase genes. Two of these new halogenase genes were found in one gene cluster of the *Botryococcus braunii* symbiont *Sphingomonas* sp. In vitro activity tests revealed that both heterologously expressed enzymes are active flavin-dependent halogenases able to halogenate indole and indole derivatives, as well as phenol derivatives, while preferring bromination over chlorination. Interestingly, SpH1 catalyses only monohalogenation, while SpH2 can catalyse both mono- and dihalogenation for some substrates.

**Keywords:** enzyme identification; flavin-dependent halogenases; bromination; metagenome screening; bacterial consortia of *Botryococcus braunii*; bioinformatics

## 1. Introduction

Halogenated organic compounds often show higher biological activities than the non-halogenated correlates. Therefore, chemical halogenation is an important methodology, but often lacks regioselectivity. Moreover, it requires relatively harsh reaction conditions and Lewis acid catalysts, and is usually performed in an organic solvent [1]. Hence, a regioselective and facile halogenation method would provide a more environmentally friendly alternative. Nature has evolved enzymatic halogenation, making use of different cofactors [2]. The enzymatic halogenation by flavin-dependent halogenases may overcome the drawbacks of chemical halogenation, since it often is highly regioselective and only requires a halide salt as a halogen source, water, oxygen, and reduced flavin–adenosine–dinucleotide ($FADH_2$) as a cofactor [1,3–6]. In general, flavin-dependent halogenases belong to the superfamily of flavin-dependent monooxygenases, which are able to activate molecular oxygen by using reduced flavin ($FADH_2$) [7], thus allowing diverse reactions, such as hydroxylation, epoxidation, and Baeyer–Villiger oxidation [8].

Tryptophan (Trp) halogenases are flavin-dependent halogenases, categorized according to their specific position of halogenating tryptophan. Trp halogenases have already been employed for halogenation on a gram scale, making use of enzyme immobilisation as cross-linked enzyme aggregates (CLEAs) [1]. The enzymatic halogenation approach has been combined with Suzuki–Miyaura cross-coupling [4,9–12], Sonogashira–Hagihara cross-coupling [13], and Mizoroki–Heck cross-coupling [14,15] in reaction cascades [4,10,11]. The tryptophan 7-halogenases PrnA, RebH, and KtzQ halogenate tryptophan regioselectively in the C7 position; the tryptophan 6-halogenases Thal, SttH, and Th-Hal prefer the C6 position, while the tryptophan 5-halogenases PyrH and ClaH modify the C5 position [16–24]. It has also been shown that it is possible to switch regioselectivity by exchanging amino acid residues. Following this strategy, Moritzer et al. redirected the regioselectivity of the tryptophan 6-halogenase Thal to the C7 position [25]. A combination of directed evolution, rational design, as well as site-saturation mutagenesis was also implemented for the design of a thermostable Thal variant with stronger elevated activity [26].

All known flavin-dependent halogenases require $FADH_2$ as a cofactor, whereas other flavin derivatives like riboflavin and flavin-mononucleotide (FMN) are not accepted [27]. $FADH_2$ is bound in the flavin binding site (motif GxGxxG) of the enzymes and reacts with oxygen to give a flavin hydroperoxide (FAD–OOH). This is followed by a nucleophilic attack of a halide forming hypohalous acid (HOX). The substrate binding site is positioned far away from the flavin binding site, but both sites are connected by a 10 Å long tunnel [28–30]. HOX passes through this tunnel, as verified by molecular dynamics calculations [31]. Different amino acid residues in the substrate binding site, as well in the tunnel, either interact with HOX or the substrate, while others are responsible for positioning the substrate to effect regioselective halogenation. Yeh et al. [32] elucidated the important role of a conserved lysine residue, e.g., K79 (PrnA) [28], K79 (RebH) [32], K83 (BrvH) [33], K74 (RadH) [34] that supposedly interacts with HOX. The potential formation of a long-lived chloramine (Lys–$N^\epsilon$H–Cl) is still under debate [32]. Alternatively, the conserved lysine residue has been postulated to stabilize and position HOX by a hydrogen bond [35]. In both cases, the electrophilic HOX species reacts with the substrate bound to the substrate binding site. This leads to a Wheland intermediate in the course of electrophilic aromatic substitution ($S_E$Ar), which is stabilized by highly conserved glutamic acid (E346 (PrnA) [28]; E357 (RebH) [36]). The carboxylate group of the conserved glutamic acid residue interacts with HOX or the chloramine, which increases electrophilicity and aligns it for regioselective halogenation. Flavin-dependent halogenases with substrates other than tryptophan do not possess the glutamic acid residue [37]. Other compounds, such as phenols or pyrroles, might be more susceptible for halogenation by HOX or the chloramine species [35]. The conserved amino acid motif WxWxIP is located at the flavin binding domain, and is believed to block binding of the substrate near the flavin cofactor, which suppresses monooxygenase activity of the enzymes [28,29,38].

Over the last years, several new flavin-dependent halogenases have been identified and heterologously expressed in different host organisms, leading to integration of halogenated substrates in the biosynthetic production of host compounds [39–42]. PrnA was the first flavin-dependent halogenase to be identified in 1997, within the pyrrolnitrin gene cluster [42]. The tryptophan 7-halogenase RebH described in 2002 is responsible for the chlorination of the secondary metabolite rebeccamycin [18]. By employing degenerative primers for highly conserved motifs, Zehner et al. identified PyrH [23], and Fujimori et al. identified KtzR and KtzQ [19] in different bacterial genomes, while Smith et al. retrieved KrmI from a marine sponge metagenome [43]. With the advent of next-generation sequencing methods, many genomes and associated metagenomes have been sequenced, and the obtained data led to the identification of many more halogenases, elucidated based on these conserved amino acid regions [22,24,44–51]. Neubauer et al. created in 2018 a profile hidden Markov model (pHMM), which led to the identification of several putative, flavin-dependent halogenases in different metagenomic data sets [33]. One of them, BrvH, was characterized with respect to halide and substrate preference in vitro, leading to the

conclusion that it halogenates indole and indole derivatives, as well as phenol derivatives, and prefers bromination over chlorination. BrvH was crystallized and revealed a structure similar to Trp halogenases, in addition to an open substrate binding site, which might lead to the acceptance of larger substrates, such as peptides [33,52]. BrvH was the first reported flavin-dependent halogenase that accepts chloride and bromide with a preference for bromination over chlorination. Only the phenol halogenases Bmp2 and Bmp5 had been reported as brominases that do not accept chloride [46]. Ismail et al. identified three novel, flavin-dependent halogenases from *Xanthomonas campestris* that exclusively brominate the substrates indole, 7-azaindole, 5-hydroxytryptophan, tryptophol, and other heterocyclic derivatives, even in the presence of an excess of chloride [51]. Like for BrvH, the crystal structure of Xcc4156 from *Xanthomonas campestris* showed an open substrate binding site. However, crystallization with FAD and bromide resulted in disruption of the crystal, which leads to the conclusion that this binding leads to positive cooperativity and conformational change [53].

Within the present work, the metagenomes of *Botryococcus braunii* communities were screened for the presence of putative, flavin-dependent halogenases by applying the generated profile hidden Markov model (pHMM) [33]. Two of the identified novel halogenases were heterologously expressed in *E. coli* and analysed in detail.

## 2. Results and Discussion

### 2.1. Identification and Analysis of Novel, Flavin-Dependent Halogenases from the Botryococcus braunii Consortia

Since the utilization of hydrocarbons by microorganisms relies on a set of different monooxygenases [54], we considered the microbial consortia accompanying the hydrocarbon-secreting microalga *Botryococcus braunii* as a potential source for novel monooxygenases [55] and potential halogenases. *Botryococcus braunii* is a colony-forming green microalga belonging to the class *Treboxiophyceae*, which can be sub-divided into distinct races depending on the type of hydrocarbon synthesized [56]. *Botryococcus* readily releases large amounts of organic carbon into the extracellular medium [57], creating a phycosphere that naturally attracts many microorganisms, including various taxa known to utilize hydrocarbons [58].

We screened metagenomic data sets of *Botryococcus braunii* consortia [59] for novel, flavin-dependent halogenases by applying our pHMM strategy [33], which is based on the PFAM database (http://pfam.xfam.org/, accessed on 8 April 2021) tryptophan–halogenase model (Trp_halogenase, PF04820), to increase the scope of possible halogenases. The metagenomes of four *Botryococcus braunii* consortia were analysed; they contained at least 33 distinct bacterial species, as indicated by 16 S rDNA amplicon and metagenome sequencing analyses [59]. For the analysis of the metagenomes, according to Neubauer et al., the pHMM was employed, which was based on Trp halogenases and was optimised in a two-step approach [33]. The analysis led to the identification of 18 complete and seven partial putative flavin-dependent halogenase genes, which possess the known conserved amino acid regions of Trp halogenases. With one exception, all halogenases found were encoded in the genomes of alphaproteobacteria, while 16 belonged to the sphingomonads genera, such as *Sphingomonas* and *Sphingopyxis* (Table 1).

A phylogenetic tree was constructed based on amino acid sequence to further analyse the identified hits and to compare these to known flavin-dependent halogenases (Figure 1). The phylogenetic analysis was conducted in MEGAX [60] by using the neighbour-joining (NJ) method and a bootstrap of 1000 [61,62]. The phylogenetic tree is divided into five groups of flavin-dependent halogenases. The phenol, pyrrol, and Trp halogenases build their own clades, respectively.

**Table 1.** Newly found putative, flavin-dependent halogenases with the conserved amino acid motif of Trp halogenases and their taxonomic assignment, based on BLAST analyses (https://blast.ncbi.nlm.nih.gov; Basic Local Alignment Search Tool).

| No. | Length (bp) | Complete | Contig Location | Phyla | Putative Origin |
|---|---|---|---|---|---|
| 1 | 225 | no | contig-391000007_2 | *Alphaproteobacteria* | *Porphyrobacter* |
| 2 | 394 | no | contig-1913000010_2 | *Alphaproteobacteria* | *Sphingopyxis* |
| 3 | 348 | no | contig-565000012_1 | *Alphaproteobacteria* | *Caulobacter* |
| 4 | 272 | no | contig-1532000014_2 | *Alphaproteobacteria* | *Sphingopyxis* |
| 5 | 503 | yes | contig-1000028_60 | *Alphaproteobacteria* | *Sphingomonas* |
| 6 | 502 | yes | contig-1029000036_1 | *Alphaproteobacteria* | *Sphingopyxis* |
| 7 | 501 | yes | contig-3201000046_41 | *Gammaproteobacteria* | *Stenotrophomonas* |
| 8 | 511 | yes | contig-2000047_22 | *Alphaproteobacteria* | *Sphingomonas* |
| 9 | 501 | yes | contig-2000047_33 | *Alphaproteobacteria* | *Sphingomonas* |
| 10 | 533 | yes | contig-2000047_110 | *Alphaproteobacteria* | *Sphingomonas* |
| 11 | 505 | yes | contig-2000047_112 | *Alphaproteobacteria* | *Sphingomonas* |
| 12 | 501 | yes | contig-2000047_113 | *Alphaproteobacteria* | *Sphingomonas* |
| 13 | 418 | no | contig-1419000070_1 | *Alphaproteobacteria* | *Caulobacter* |
| 14 | 359 | no | contig-832000073_1 | *Alphaproteobacteria* | *Porphyrobacter* |
| 15 | 501 | yes | contig-252000084_3 | *Alphaproteobacteria* | *Sphingopyxis* |
| 16 | 501 | yes | contig-2000086_90 | *Alphaproteobacteria* | *Brevundimonas* |
| 17 | 443 | no | contig-2264000090_1 | *Alphaproteobacteria* | *Sphingopyxis* |
| 18 | 521 | yes | contig-16000092_39 | *Alphaproteobacteria* | *Sphingomonas* |
| 19 | 502 | yes | contig-867000143_2 | *Alphaproteobacteria* | *Caulobacter* |
| 20 | 513 | yes | contig-3212000146_163 | *Alphaproteobacteria* | *Sphingomonas* |
| 21 | 512 | yes | contig-3212000146_164 | *Alphaproteobacteria* | *Sphingomonas* |
| 22 | 553 | yes | contig-3212000146_184 | *Alphaproteobacteria* | *Sphingomonas* |
| 23 | 493 | yes | contig-1000154_7 | *Alphaproteobacteria* | *Sphingomonas* |
| 24 | 522 | yes | contig-981000159_36 | *Alphaproteobacteria* | *Brevundimonas* |
| 25 | 516 | yes | contig-981000159_37 | *Alphaproteobacteria* | *Brevundimonas* |

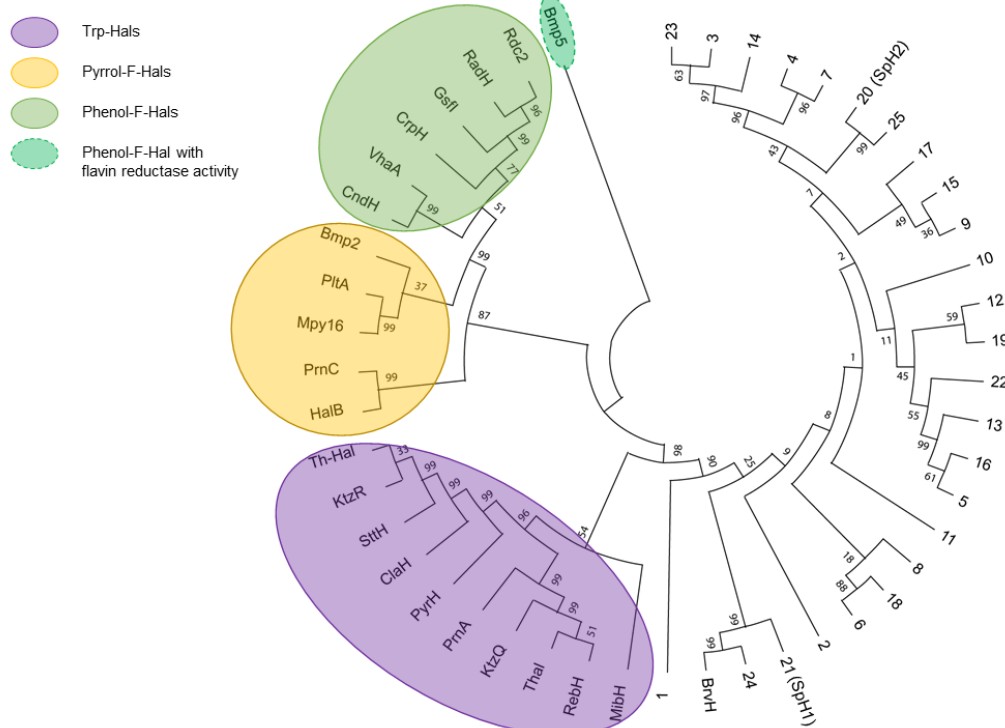

**Figure 1.** Phylogenetic tree of the hits from the *Botryococcus braunii* consortia, as well as known Trp halogenases, phenol halogenases, and pyrrol halogenases. The optimal tree, with the sum of branch lengths = 17.80254621, is shown. The newly identified hits all cluster with the flavin-dependent halogenase BrvH. We decided to further analyse hit 21 (SpH1) and hit 20 (SpH2).

Noteworthy, Bmp5 is outside the phenol halogenase clade, although it is a phenol halogenase. However, it also possesses flavin reductase activity, which might be the reason for building its own clade [46]. The newly identified hits from the *B. braunii* bacterial consortia cluster close to Trp halogenases, but build their own clade together with BrvH. BrvH was found to be very similar to hits 24 and 21 (pairwise identity of 95% and 64%, respectively), which represent flavin-dependent halogenases encoded in the metagenome-assembled genomes (MAG 10 and MAG 21, respectively [59]) of *Brevundimonas* and *Sphingomonas*. Thus, these enzymes might represent their own group within the flavin-dependent halogenases with likely similar substrate preferences. According to antiSMASH analyses (antibiotics & Secondary Metabolite Analysis Shell [63]), both MAGs encode several metabolic gene clusters for bacteriocins and antibiotic compounds [59,64]. In addition, the MAG similar to *Sphingomonas* contains a gene cluster potentially responsible for siderophore synthesis.

We decided to investigate the identified gene 21 (J) from *Sphingomonas*. However, during the gene cluster analysis, the presence of another flavin-dependent halogenase gene (20, I) located upstream to 21 (J) was detected (Figure 2). Some of the known flavin-dependent halogenases have been reported to originate from the same gene cluster, e.g., PrnA and PrnC that play a role in pyrrolnitrin biosynthesis [42]. Bmp2 and Bmp5 likewise halogenate different building blocks in the biosynthesis of natural, polybrominated marine products [41]. KtzR and KtzQ act together in the biosynthesis of kutzneride. In this case, KtzQ is responsible for the chlorination of tryptophan at C7, while KtzR catalyses the second halogenation to give 6,7-dichlorotryptophan. KtzR also catalyses chlorination of tryptophan but possesses a higher affinity to 7-chlorotryptophan [19].

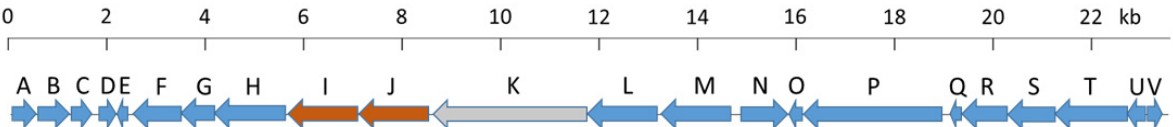

A: Catabolite regulation protein A (CreA)
B: Acyltransferase
C: Small integral membrane protein
D: Hypothetical protein
E: DUF1059 domain-containing protein
F: Sensor histidine kinase
G: Phosphate regulatory protein PhoB
H: Membrane protein DUF4153
I: Tryptophan halogenase superfamily (SpH2)
J: Tryptophan halogenase superfamily (SpH1)
K: Catecholate siderophore receptor

L: Glycosyl hydrolase family
M: NAD-dependent dehydrogenase
N: Cyn operon transcriptional activator
O: Hypothetical protein
P: Protease
Q: ABC transporter substrate-binding protein
R: ABC transporter permease
S: ABC transporter permease
T: Sugar ABC transporter ATP-binding protein
U: Carbohydrate kinase
V: Sensory transduction protein LytR

**Figure 2.** Gene cluster in the vicinity of hit 21 (J; SpH1) in the metagenome-assembled genome (MAG) 21 of the *B. braunii* bacterial consortia. Another flavin-dependent halogenase gene, *sph2* (J), was identified upstream to *sph1* (I). The genes I, J, K, and L were found to be encoded on the same contig in the metagenomics dataset.

Since many flavin-dependent halogenases originating from the same gene cluster were reported to act in concert [17,19,46], both putative, flavin-dependent halogenases 21 (J) and 20 (I)—in the following named SpH1 (21) and SpH2 (20)—were heterologously expressed in *E. coli* and subjected to further analyses. Both enzymes possess the typical conserved amino acid regions of flavin-dependent halogenases like the flavin binding domain and the conserved lysine residues. However, both show 42% pairwise identity to each other, while the pairwise identity to BrvH was 64% for SpH1 and 42% for SpH2. The amino acid sequences are provided in the Supplementary Materials.

### 2.2. In Vitro Experiments with the Flavin-Dependent Halogenases SpH1 and SpH2

2.2.1. Determination of Halogenation Activity and Substrate Scope

In order to confirm that the identified genes are active halogenating enzymes, experiments with the novel enzymes were conducted in vitro. The genes *sph1* and *sph2* were heterologously expressed in *E. coli*, together with the chaperone system GroEL/GroES. Both enzymes were obtained in soluble form in high concentrations (Supplementary Materials). For the substrate screening, the previously established reaction conditions, with a cofactor regeneration by flavin reductase (PrnF), and *Rhodococcus ruber* alcohol dehydrogenase (*RR*-ADH), were used [3]. Indole and L-tryptophan were the first tested substrates, because SpH1 and SpH2 are similar to BrvH in brominating indole [33]. Similar to the activity observed for BrvH, SpH1 and SpH2 catalyse the bromination of indole, but not L-tryptophan (Figure 3), thus revealing that both novel identified enzymes represent active halogenases.

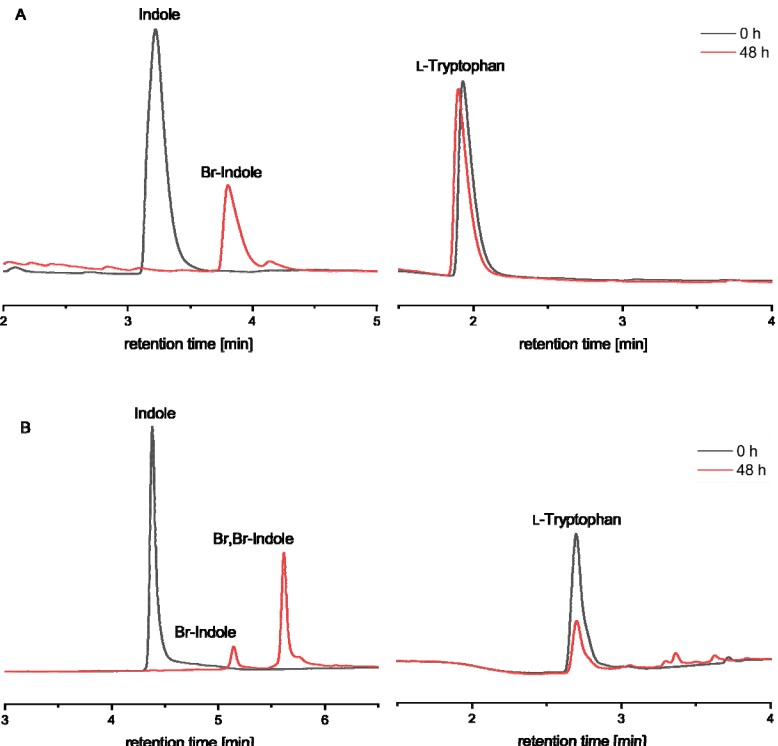

**Figure 3.** Reverse-phase high-performance liquid chromatography (RP-HPLC) profiles (detection at 280 nm) of the conversions of 1 mM indole or 1 mM L-tryptophan by SpH1 (**A**) and SpH2 (**B**) in presence of 100 mM NaBr as a halide source. *Rhodococcus ruber* alcohol dehydrogenase (*RR*-ADH) and flavin reductase PrnF were used for cofactor regeneration. Retention times differ due to different HPLC methods. (**A**) SpH1 (HPLC method 1); (**B**) SpH2 (HPLC method 2). SpH1 had completely halogenated indole (retention time $t_R$: 3.2 min) after 48 h of incubation to give bromoindole ($t_R$: 3.8 min), but L-tryptophan ($t_R$: 2 min) was not converted. SpH2 catalyses the halogenation of indole ($t_R$: 4.4 min) to give bromoindole ($t_R$: 5.1 min) and dibromoindole ($t_R$: 5.6 min), without converting L-tryptophan ($t_R$: 2.7 min).

The newly identified brominating enzymes belong to the class of flavin-dependent halogenases. Both enzymes have the characteristic flavin binding module GxGxxG, the conserved lysine residue (K85 and K81, respectively), and the WxWxIP motif suppressing monooxygenase activity [33]. The reduced flavin (FADH$_2$) reacts with molecular oxygen, forming the FAD(C4a)-peroxide, which in turn reacts with halide ions under HOX formation [2,5,28,29]. In contrast, heme- or vanadium-dependent haloperoxidases require hydrogen peroxide and release the halogenating species hypohalous acid HOX into the medium, which leads to unselective halogenation [2,65,66]. Flavin-dependent halogenases

can be distinguished from haloperoxidases by the monochlorodimedon assay [67]. In this assay, monochlorodimedon reacts with free HOX released by the haloperoxidase to give rise to a dihalogenated derivative, which can be monitored photometrically. This test was negative for both enzymes. The addition of catalase, which decomposes hydrogen peroxide and would hence stop the halogenation reaction by haloperoxidases, did not negatively affect the halogenation reaction, leading to the conclusion that SpH1 and SpH2 are not haloperoxidases (Supplementary Materials). FAD reconstitution is another possibility for the classification towards flavin-dependent halogenases [33,68]. SpH1 and SpH2 were incubated with FAD overnight, and then the buffer was changed to a buffer without FAD, leading to no free FAD in the solution. UV/Vis spectroscopy can be employed for the detection of bound FAD in the enzymes. For both SpH1 and SpH2, the absorption band at 350 nm shifted to 348 nm, and the absorption band at 373 nm shifted to 360 nm, showing that both enzymes bind FAD and can therefore be classified as flavin-dependent halogenases (Supplementary Materials). The SpH1 mutant where the conserved lysine residue K85 was replaced by an alanine (SpH1_K85A; Supplementary Materials) was completely inactive. Hence, halogenase SpH1 depends on the lysine, which is known to be essential for the flavin-dependent halogenases.

A pharmacophore model was adapted from a virtual screening methodology for BrvH, and revealed 19 compounds halogenated by BrvH [52]. SpH1 is highly similar to BrvH and clusters within one clade with BrvH. Therefore, its substrate scope might be similar as well. In addition to indole and L-tryptophan, further substrates, such as tryptophan and indole derivatives, as well as other aromatic compounds, were tested for halogenation by SpH1 and SpH2 (Figure 4). Twenty of the 26 accepted substrates represent indole derivatives, but phenol derivatives (i.e., phenol (**24**), anthranilic acid (**25**), and 4-*n*-hexylresorcinol (**18**)), azulene (**17**), and quinoxaline (**22**) were also accepted. Enzymatic conversion was verified by reverse-phase high-performance liquid chromatography (RP-HPLC), as well as mass analysis (Supplementary Materials).

SpH1 and SpH2 were immobilised as CLEAs to convert higher amounts of substrate, in order to identify their regioselectivity. The halogenases and cofactor-regenerating enzymes were immobilised together using glutaraldehyde, and the CLEAs were incubated for 10 to 14 days with substrate. The obtained product was purified and analysed by NMR spectroscopy. Mass spectrometry (MS) analyses revealed the correct isotopic pattern of the brominated products (Supplementary Materials). All indole derivatives analysed so far are halogenated at C3, if this position is unsubstituted. Only when a non-hydrogen substituent is present at C3, halogenation occurs at C2 (Figure 4). The C3 and C2 positions are electronically the most favored positions for electrophilic substitution within the indole ring [69]. Likewise, PrnA and RebH halogenate indole and some derivatives in the most activated position. The regioselectivity strongly depends on the substrate and its position in the active site [3,70]. SpH2 is also able to halogenate 2,3-methylindole. In this case, it halogenates the substrate at C6, but also a minor side product halogenated at C5 was observed, leading to the conclusion that SpH2 is not strictly regiospecific for 2,3-methylindole.

Furthermore, SpH1 and SpH2 show higher bromination activity compared to chlorination activity. They brominated 1 mM indole to 100% within 48 h, while SpH1 under similar conditions chlorinated only to 10%, and SpH2 to 17%. Interestingly, dichlorinated product was not found for SpH2. The brominated, as well as the chlorinated products, were verified via ESI-MS or liquid chromatography LC-MS analyses (**SpH1:** positive mode: $[C_8H_6BrN]^+$, calcd. 195.97 ($^{79}$Br), 197.97 ($^{81}$Br), obs. $[M+H]^+$ 195.55/197.53; negative mode: $[C_8H_6ClN]^-$, calcd. 150.02 ($^{35}$Cl), 152.02 ($^{37}$Cl), obs. $[M-H]^-$ 150.01/152.01; **SpH2:** negative mode: $[C_8H_6BrN]^-$, calcd. 193.97 ($^{79}$Br), 195.97 ($^{81}$Br), obs. $[M-H]^-$ 193.96/195.95; $[C_8H_6ClN]^-$, calcd. 150.02 ($^{35}$Cl), 152.02 ($^{37}$Cl), obs. $[M-H]^-$ 150.01/152.01).

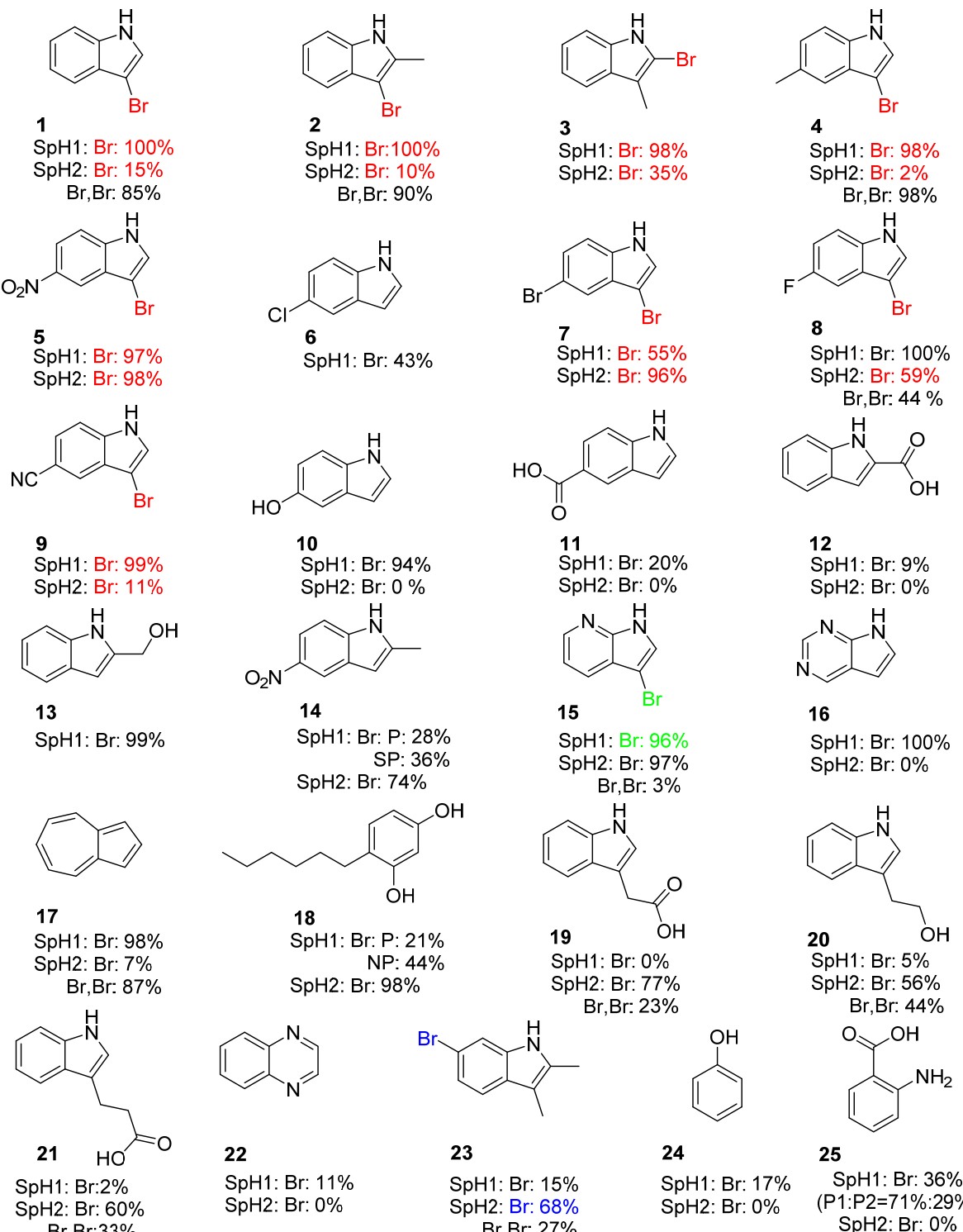

**Figure 4.** Halogenated substrates by SpH1 and SpH2. The conversions were calculated by determining the ratio of the peak areas of substrates and products. The position of halogenation was determined for 10 substrates by NMR analysis (red: SpH1 and SpH2, green: SpH1, and blue: SpH2). SP: different unidentified side products; P1, P2: two monobrominated products at a ratio of P1/P2.; NP: side product.

SpH1 and SpH2 are thus similar to BrvH and the Xcc halogenases Xcc4156, Xcc1333, and Xcc4345, which all prefer bromination over chlorination [33,51]. Noteworthy, the Xcc halogenases, SpH1, and SpH2 originate from terrestrial habitats, but favour bromination over chlorination. The Xcc halogenases were identified in *Xanthomonas campestris*, which

is a plant pathogen [51], and SpH1 and SpH2 originate from *B. braunii* consortia, which mainly inhabit fresh water [57]. Usually, it is presumed that because of the higher bromide concentration in sea water, bromination is preferred in marine habitats, while halogenases from terrestrial habitat would favour chlorination [71].

In 2019, the Lewis group identified many flavin-dependent halogenases via family-wide activity profiling, and most of them preferred bromination over chlorination [72]. The group postulated that bromination activity is more widespread than chlorination activity of flavin-dependent halogenases, which matches our data. However, even RebH, which chlorinates its natural substrate tryptophan, was shown in competition experiments to prefer bromination [72]. Therefore, it is also possible that SpH1 and SpH2 chlorinate their natural substrate.

However, the natural substrates of SpH1 and SpH2 are yet unknown. SpH2 accepts larger substrates, such as (3-indolyl)acetic acid (**19**), 3-(3-indolyl)propionic acid (**21**), and tryptophol (**20**). This possible substrate group could be represented by auxins, pivotal plant hormones (also called phytohormones), with (3-indolyl)acetic acid (IAA) (**19**) as one of the best characterised class of regulators [73,74]. The chlorinated form of auxin, (4-chloro-3-indolyl)acetic acid (4-Cl-IAA), is a highly active hormone that is thought to play a key role in early pericarp growth [75] and fruit development in peas [76]. Auxins also represent a key modulator of plant–bacteria interactions with the bacterial ability to synthesise, e.g., IAA being an attribute for both promoting plant growth and phytopathogenic effects [77]. So far, however, there is no evidence in the literature about brominated auxins in nature, although the chemical synthetic bromination of (3-indolyl)acetic acid has been reported [78]. Another interesting hint towards possible natural substrates is represented by the gene located downstream next to *sph1* (Hit 21; J), which encodes for a catecholate siderophore receptor (Figure 2). Since the MAG similar to *Sphingomonas* also contains at least one metabolic gene cluster potentially responsible for siderophore synthesis (Supplementary Materials), siderophores may represent another potential substrate for the halogenases SpH1 and SpH2. Siderophores possess phenol moieties, which might be halogenated by SpH1 and SpH2, since it was shown that these are able to halogenate 4-*n*-hexylresorcinol (**18**), phenol (**24**), and anthranilic acid (**25**) (Figure 4). In addition to iron scavenging, siderophores can have many alternative functions, including non-iron-metal transport, signaling, and antibiotic activity functions [79]. The halogenation of siderophores is common in nature; for instance, two *Streptomyces* siderophores, chlorocatechelins A and B, have been reported to contain chlorinated catecholate groups [80].

### 2.2.2. Investigation of Mono- and Dibromination Activity

Interestingly, SpH1 only catalyses monobromination, while SpH2 is able to catalyse both mono- and dibromination of indole and its derivatives (Figure 4). Kinetic measurements showed that SpH2 has a negligibly higher specific activity towards indole ($4.2 \pm 0.03$ mU/mg) than towards 3-bromoindole ($3.8 \pm 0.45$ mU/mg). In comparison, the specific activity of SpH1 towards indole is $3.0 \pm 0.7$ mU/mg. The synergistic cooperation of two enzymes in the halogenation process is known in the literature [17,19,46]. PrnA and PrnC have been identified as catalysing the chlorination of tryptophan and monodechloroaminopyrrolnitrin in the biosynthesis cluster of pyrrolnitrin. PrnC catalyses the chlorination of monodechloroaminopyrrolnitrin, halogenated by PrnA in the first step [17,28]. Another example is the biosynthesis of kutzneride, which includes KtzQ and KtzR [19].

SpH1 and SpH2 were evaluated in silico for unveiling the dibrominating ability exhibited by SpH2. Since no crystal structure was available for either of the two enzymes, the homology model was built using YASARA structure (www.yasara.org; 2020) for in silico study of the novel halogenases. SpH2 exhibited a wider active site pocket with an area of 136.8 Å2, compared to SpH1 with 77.39 Å2 (Figure 5 and Supplementary Materials).

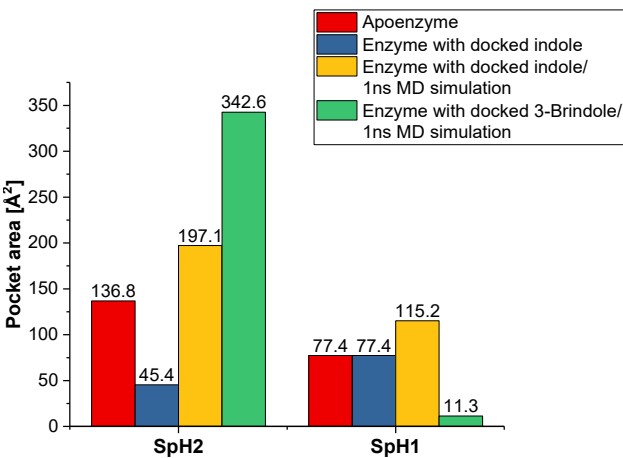

**Figure 5.** Active site pocket area for the apo-, indole-, and 3-bromoindole-docked SpH2 and SpH1 halogenases, showing the flexibility of SpH2 in accommodating 3-bromoindole, unlike SpH1.

The active site pockets of both halogenases differ in their geometry, as does their depth within the pocket for reaching the conserved amino acid residues (Lys and Glu). The SpH1 active site pocket is deeper compared to SpH2, with the loop containing the conserved glutamic acid residue (Glu348) being displaced backwards. On the other hand, the SpH2 active site pocket is wider and not as deep as that of SpH1; still, the pocket is extended to allow the substrate to reach the conserved Lys residue ( Figures 6 and 7, and Supplementary Materials).

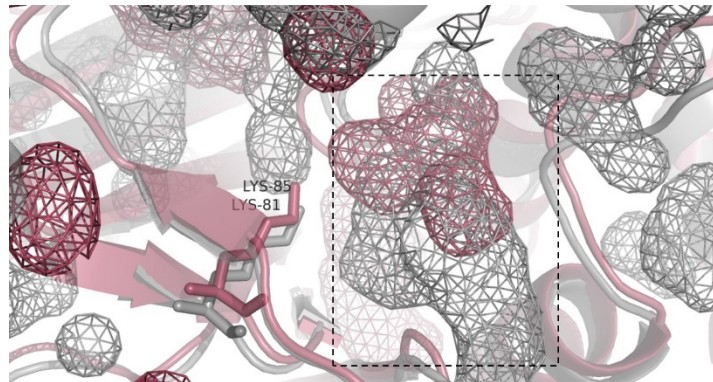

**Figure 6.** Superimposed homology model of SpH2 (grey) and SpH1 (red) after 1ns molecular dynamics (MD) simulation, showing the active site cavity in wireframe (in dashed-line box). The conserved active site lysine residues 81 and 85 in SpH2 and SpH1, respectively, are shown.

Indole was docked into both models to indicate its flexibility and expansion within the active site pocket of SpH2. The docked substrates were then subjected to MD simulation to further understand the changes that might occur as a result of indole binding to the active site. Obvious differences were detected in the size of the active site pockets containing indole after MD simulation, where the SpH2 active site was expanded to allow the binding of bulkier substrates, thus showing that the active site is flexible. However, SpH1 was not largely expanded, indicating less flexibility to accommodate larger substrates. Both cases showed H-bond formation with the indole NH group (with Ser445 in SpH1 and Ser442 in SpH2; Figures 5 and 7, and Supplementary Materials). Furthermore, the binding affinity of the docked indole towards both enzymes revealed a slight preference for SpH2 over SpH1 (Table 2), which corresponds to the specific activity calculated for SpH2 towards indole compared to SpH1.

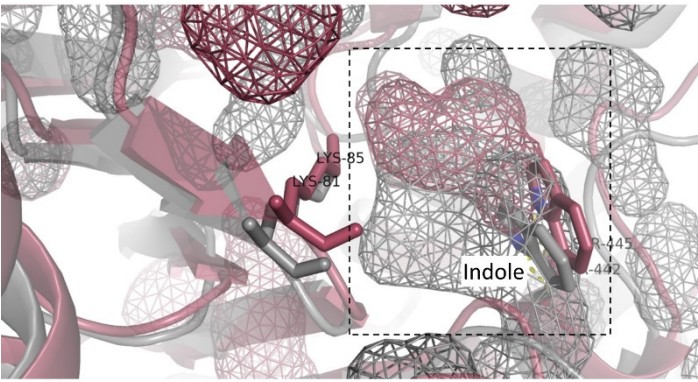

**Figure 7.** Superimposed homology model of SpH2 (grey) and SpH1 (red) with the docked indole after 1 ns MD simulation. The active site cavity is shown in wireframe (surrounded by dashed-line box). In both cases, an H-bond formed between indole NH and Ser442 and Ser445 in SpH2 and SpH1, respectively.

**Table 2.** Binding affinity of indole and 3-bromoindole to the active site of SpH2 and SpH1.

| Enzyme/Substrate | Mode | Binding Affinity [kJ/mol] |
|------------------|------|---------------------------|
| SpH2/indole | 3 | 22.9 |
| SpH1/indole | 9 | 20.1 |
| SpH2/3-bromoindole | 5 | 22.5 |
| SpH1/3-bromoindole | 11 | 19.4 |

In addition, 3-bromoindole was docked into the homology models. Higher affinity towards 3-bromoindole was detected for SpH2 over SpH1, which provides further evidence for the dibromination ability of SpH2 (Table 2). In addition, the size of the active site pocket in the case of SpH2 was significantly expanded compared to the active site pocket of SpH1 (Figure 5 and Figure SI9), which is more rigid, and hence would prohibit further bromination of the 3-bromoindole (Figure 8).

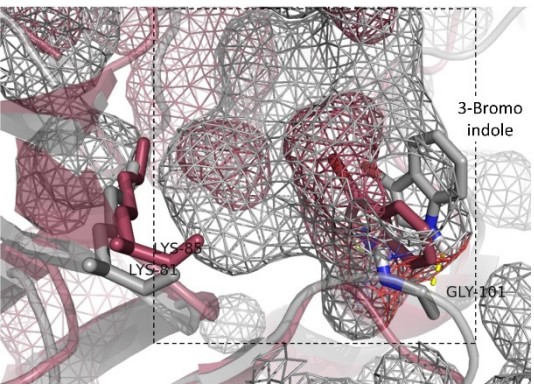

**Figure 8.** Superimposed homology model of SpH2 (grey) and SpH1 (red) with the docked 3-bromoindole after 1 ns MD simulation. The active site cavity is shown in wireframe (surrounded by a dashed-line box). In both cases, an H-bond is formed between indole NH and Gly442 in SpH2 and Ser445 in SpH1.

The in silico simulation of the SpH1 and SpH2 homology models underlines our findings that SpH2 is able to catalyse dihalogenation. Both SpH1 and SpH2 originate from one gene cluster, and therefore could act in concert and interact with each other. SpH1 has a deeper but less flexible active site pocket, which is likely to be suitable for the binding of smaller substrates like indole, while SpH2 features a wider and more flexible active site

pocket, likely accommodating bulkier substrates, as well as the dibromination observed with some substrates.

## 3. Materials and Methods

Solvents and chemicals were, unless otherwise specified, of analytical grade (p.a.) and purchased from commercial suppliers. Chemicals such as *iso*-propanol, acetonitrile, sodium chloride/bromide, ethanol, and methanol were purchased from VWR Chemicals (Darmstadt, Germany) and L-arabinose, NADH/NAD, FAD and glutaraldehyde from Carl Roth (Karlsruhe, Germany). Antibiotics were obtained from Fisher Scientific (Waltham, MA, USA; ampicillin sodium salt), AppliChem (Darmstadt, Germany; chloramphenicol) and Carl Roth (Karlsruhe, Germany; kanamycin sulfate).

Chemicals for substrate assays were acquired at Acros Organics (Fair Lawn, NJ, USA; indole, indazole, 5-cyanoindole, 3-(3-indolyl)propionic acid, indole-3-carbaldehyde, trypthophan, tryptamine, tryptophol), Sigma Aldrich (Darmstadt, Germany; 2-methylindole, 4-hydroxybenzoic acid, 5-aminoindole, 5-bromoindole, indole-2-carboxylic acid, phenylalanine, phenol, 7*H*-pyrolo[2,3-d]pyrimidine, quinoxaline), Alfa Aesar (Kandel, Germany; 2,3-dimethylindole, 3-methylindole, 4-*n*-hexylresorcinol, 5-hydroxyindole, 5-fluoroindole, 5-methylindole, azulene, indole-3-acetonitrile, (3-indolyl)acetic acid, indole-5-carbaldehyde, indole-5-carboxylic acid, 5-hydroxytryptophan hydrate, naphthylacetic acid), Maybridge (Altrincham, UK; 5-nitroindole), Carl Roth (Karlsruhe, Germany; 7-azaindole) and Fluorochem (Derbyshire, UK; benzoxazole).

The plasmid pClBhis–prnF encoding the flavin reductase PrnF from *Pseudomonas fluorescens*, was donated by Prof. Dr. Karl-Heinz van Pée, Technical University Dresden, and the plasmid vector pET-21–adh encoding the alcohol dehydrogenase from *Rhodococcus ruber* (*RR*-ADH) was kindly provided by Prof. Dr. Werner Hummel, Bielefeld University. The plasmid pGro7 coding for the chaperone system GroEL–GroES was purchased from TaKaRa Bio Inc. *E. coli* DH5 and *E. coli* BL21 (DE3) were purchased from Novagen.

### 3.1. Analytics

#### 3.1.1. Analytical, Reverse-Phase High-Performance Liquid Chromatography (RP-HPLC)

For analytical, reverse-phase high performance liquid chromatography, two different methods were employed.

With method 1, reactions were monitored using a Thermo Scientific (Waltham, MA, USA) Accela 600 with NUCLEOSHELL RP 18 column 18.5 μm from Macherey-Nagel (Düren, Germany; 150 × 2.1 mm, eluent A: $H_2O/CH_3CN/TFA$ = 95.0:5.0:0.1; eluent B: $H_2O/CH_3CN/TFA$ = 5.0:95.0:0.1, flow rate 900 μl/min,; 0–1 min 100% eluent A, 1–6 min linear gradient to 100% eluent B, 6–7 min 0% eluent A, 7–7.5 min back to 100% eluent A, 7.5–9 min 100% eluent A).

With method 2 for analytical RP-HPLC analysis a Shimadzu (Duisburg, Germany) Nexera XR chromatography system was employed (Luna C18(2) column (3 μM, 100 Å, LC column, 100 × 2 mm) from Phenomenex (Aschaffenburg, Germany; eluent A: $H_2O/CH_3CN/TFA$ = 95.0:5.0:0.1; eluent B: $H_2O/CH_3CN/TFA$ = 5.0:95.0:0.1; flow rate 650 μL/min; 0–5.5 min linear gradient to 95% eluent B, 5.5–6.0 min 95% eluent B, 6.0–6.1 min back to 95% eluent A, 6.1–9.0 min 95% eluent A).

#### 3.1.2. Preparative, Reverse-Phase High-Performance Liquid Chromatography (RP-HPLC)

For the isolation and purification of products, a Merck-Hitachi (Darmstadt, Germany) LaChrom RP-HPLC system was employed. Separation of the products was conducted using a Hypersil Gold C18 column (8 μM, 250 × 21.2 mm) from Thermo Scientific (Waltham, MA, USA). Absorbance was measured simultaneously at 220 nm, 254 nm, and 280 nm. The setup was as follows: eluent A, $H_2O/CH_3CN/TFA$ = 95.0:5.0:0.1; eluent B, $H_2O/CH_3CN/TFA$ = 5.0:95.0:0.1; flow rate = 10 mL/min; 0–45 min linear gradient to 100% eluent B; 45–50 min 100% eluent B; 50–55 min 100% eluent A.

### 3.1.3. High-Performance Liquid Chromatography–Mass Spectrometry (HPLC-MS)

For the identification of products by electrospray ionisation, a high-performance liquid chromatography–mass spectrometry (HPLC-MS) system consisting of an Agilent 1200 HPLC system and a 6220 TOF mass spectrometer (Agilent Technologies, Santa Clara, CA, USA) was used, with the following setup: solvent A, $H_2O/CH_3CN/FA = 95.0:5.0:0.1$; solvent B, 5.0:95.0:0.1; flow rate = 0.3 mL/min, with a linear gradient from 0% to 98% B over 10 min, 1 min at 98% B, and back to 0% B for 5 min.

### 3.1.4. Gas Chromatography–Mass Spectrometry (GC-MS)

For gas chromatography–mass spectrometry (GC-MS) analysis, a system consisting of a Trace GC Ultra gas chromatograph (Thermo Scientific, Waltham, MA, USA) and an ITG900 mass spectrometer from Thermo Finnigan (20 measurements per minute, $50 \pm 750$ *m/z*) was applied. Separation of the products was conducted using a VF-5 column (0.25 µM, 30 m × 0.25 mm, 5% diphenylsiloxan, 95% dimethylsiloxan) from Thermo Scientific (Waltham, MA, USA). Helium was used as the mobile phase, with a temperature gradient of 5 °C/min from 80 °C to 325 °C.

### 3.1.5. Nuclear Magnetic Resonance (NMR) Spectroscopy

NMR spectra were recorded on an Avance 500 ($^1$H: 500 MHz, $^{13}$C: 126 MHz) or a DRX-500 spectrometer ($^1$H: 500 MHz, $^{13}$C: 126 MHz) (Bruker, Billerica, MA, USA). Chemical shifts are reported relative to residual solvent peaks (DMSO-$d_6$: $^1$H: 2.5 ppm; $^{13}$C: 39.5 ppm).

### *3.2. Bioinformatic Analysis*
### 3.2.1. Metagenomic Analysis for the Detection of Flavin-Dependent Halogenases

Metagenomic datasets of *Botryococcus braunii* consortia, deposited under the BioProjectID PRJEB26344 [59], were used for the search of novel flavin-dependent halogenases. The analyses were performed according to the previously described pHMM strategy [33], which is based on the PFAM (http://pfam.xfam.org, 2016) tryptophan–halogenase model (Trp_halogenase, PF04820). The dataset of the MAG21 (similar to the genus *Sphingomonas*) can be found under the deposited BioProject ID PRJEB26345 [59], and was used for the antiSMASH (https://antismash.secondarymetabolites.org, 2020) analysis (using default settings) [63]. Amino acid sequence alignments were performed using NCBI ( www.ncbi.nlm.nih.gov, accessed on 8 April 2021, Bethesda, MD, USA, 2020) NR [81] and MultAlin [82]. The assignment of the active sites of the enzymes was accomplished on the basis of the published information for RebH and PrnA [29,32,36].

### 3.2.2. Phylogenetic Analysis

Phylogenetic analyses were performed with MEGA7 (www.megasoftware.net, accessed on 8 April 2021) [60], based on protein alignment (MUSCLE) [83], neighbour joining (NJ) method, and a bootstrap of 1000 [61,62].

### *3.3. In Vitro Enzyme Assays*
### 3.3.1. Vector Construction and Molecular Cloning

Gene synthesis of the putative halogenase genes *spH1* and *spH2* was ordered from ThermoFisher GeneArt (Waltham, MA, USA). The genes pMA-T-spH1 and pMA-T-spH2 were codon-optimised for *E. coli* and provided with restriction sites (BamHI and NdeI (New England BioLabs, Frankfurt am Main, Germany)). The synthetic genes were cloned into pET28a expression vectors and transformed in *E. coli* BL21 (DE3) pGro7.

### 3.3.2. Heterologous Protein Expression and Purification

A total of 1.5 L of *Luria-Bertani* medium, containing 60 µg/mL kanamycin and 50 µg/mL chloramphenicol, was inoculated with 2% of an overnight culture of *E. coli* BL21 (DE3) pGro7 pET28a-spH1 or pET28a-spH2, and incubated at 37 °C until reaching

an optical density ($OD_{600}$) of 0.4. The temperature was decreased to 25 °C for 30 min, and overexpression was induced by the addition of isopropyl-β-D-thiogalactopyranoside (0.1 mM) and L-arabinose (2 g/L). After cultivation for a further 22 h at 150 rpm, cells were harvested by centrifugation (4200× *g*, 1 h, 4 °C), washed with 100 mM $Na_2HPO_4$ buffer (pH 7.4), and stored at −20 °C.

Expression of alcohol dehydrogenase *RR*-ADH and flavin reductase PrnF was performed as previously described [3].

The cells from 1.5 L cultivation were resuspended in 30 mL of 100 mM $Na_2HPO_4$ buffer (pH 7.4) and lysed by French Press (three times, 1000 psig). Cell debris was removed by centrifugation (10,000× *g*, 30 min, 4 °C), and soluble protein was filtered through a 0.2 μM Whatman filter. The purification of the hexahistidin-tagged proteins was performed via a HisTALON matrix. The lysate was loaded on HisTALON agarose affinity resin (TaKaRa Bio Inc., Saint-Germain-en-Laye, France) and washed with 10× column volume (CV) with 100 mM $Na_2HPO_4$ buffer (pH 7.4), and 10 × CV 50 mM $Na_2HPO_4$ buffer (pH 7.4) with 300 mM NaBr and 10 mM imidazole. The proteins were eluted with 50 mM $Na_2HPO_4$ buffer (pH 7.4), 300 mM NaBr, and 300 mM imidazole and collected in 0.75 mL fractions. After determination of the protein concentration by NanoDrop UV spectroscopy (Thermo Scientific, Waltham, MA, USA), fractions with the purified proteins were pooled and desalted via a HiTrap Desalting column, with 50 mM $Na_2HPO_4$ buffer (pH 7.4) and 50 mM NaBr.

### 3.3.3. Enzymatic Halogenation/Enzyme Assay with Purified Protein on an Analytical Scale

The activities of PrnF and ADH were determined as published by Frese et al. [3]. The halogenation activities of SpH1 and SpH2 towards different substrates were carried out for 48 h at 25 °C and 500 rpm in a total volume of 0.5 mL, containing 1.25 mg/mL enzyme, 1 mM substrate, 1 μM FAD, 100 μM NAD, 100 mM NaBr/NaCl, *RR*-ADH (2 U/mL) PrnF (2.5 U/mL), 5% (*v/v*) *iso*-propanol, and 100 mM $Na_2HPO_4$ buffer (pH 7.4) [33,52]. Samples were quenched by adding an equal volume of methanol. Reactions were analysed by analytical RP-HPLC and HPLC-MS.

### 3.3.4. Determination of Specific Activity

Halogenation reactions, as described in Section 3.3.3., were carried out with 0.05 mM substrate and 10 μM enzyme. Samples were taken at 5 min intervals over a 20 min time period, and then the enzyme reaction was stopped with 1:1 methanol/water (*v/v*). The determination was carried out in triplicate, reactions were analysed by analytical RP-HPLC, and conversion rates (*v*) were calculated. With this in hand, the specific activity can be calculated using the following:

$$specific\ activity\ \left[\frac{mU}{mg}\right] = \frac{v}{c_E \cdot M_E} \times 10^9 \tag{1}$$

where $c_E$ is the enzyme concentration (10 μM) and $M_E$ is the enzyme mass (g/mol).

### 3.3.5. Catalase Activity Assay

Catalase from bovine liver lyophilised powder (2000–5000 units/mg protein; Sigma Aldrich, Darmstadt, Germany) at a final concentration of 100 U/mL was additionally added to the reaction mixture, and the procedure continued as described above to determine the FAD dependency of SpH1 and SpH2 [33].

### 3.3.6. Enzymatic Halogenation with Immobilised Protein on a Preparative Scale

For the isolation and analysis of halogenated products on a preparative scale, cross-linked enzyme aggregates (CLEAs), as described by Frese et al. [1], were used.

The *E. coli* BL21 (DE3) pGro7 cells from a 1.5 L cultivation containing SpH1 and SpH2 were resuspended in 30 mL 100 mM $Na_2HPO_4$ buffer (pH 7.4) and lysed by French Press

(three times, 1000 psig). After centrifugation (30 min, 10,000× *g*, 4 °C), the cell lysate was divided in two equal parts, and PrnF (2.5 U/mL) and *RR*-ADH (1 U/mL) were added. Precipitation of the proteins was carried out by adding ammonium sulfate (8.1 g, 95% saturation), followed by incubation for 1 h at 4 °C in a tube rotator. Glutaraldehyde (1.5 mL, 0.5% (*w/v*)) was added for cross-linking, and the mixture was further kept for 2 h at 4 °C. The CLEAs were centrifuged and washed three times with 30 mL of 100 mM $Na_2HPO_4$ buffer (pH 7.4). The pH of the reaction buffer for the biocatalysis with CLEAs containing, 1.5 mM substrate, 1 μM FAD, 100 μM NAD, 30 mM NaBr/NaCl, and 5% (*v/v*) and 15 mM $Na_2HPO_4$, was adjusted with phosphoric acid to pH 7.4. Finally, CLEAs were added to reaction buffer in a final volume of 500 mL, and incubated for up to 10 days (until no further conversion was visible at the analytical RP-HPLC) at 25 °C and 150 rpm. The reaction mixture was filtered and extracted three times with 250 mL dichloromethane. The organic phase was dried over magnesium sulfate and evaporated. Before the characterisation of the halogenated products with HPLC-MS, analytical HPLC, and NMR, the remaining substance was purified over preparative RP-HPLC and lyophilised.

### 3.3.7. FAD Reconstitution

For FAD reconstitution, the buffer of the purified proteins was exchanged to FAD reconstitution buffer (50 mM $Na_2HPO_4$, 150 mM NaBr, 1 mM FAD, pH 7.4) using a HiTrap-Desalting column and incubated overnight at 4 °C on a rocking shaker. The proteins were washed three times with 50 mM $Na_2HPO_4$ (pH 7.4) and 150 mM NaBr to remove free FAD, and concentrated to a final volume of 400 μL in an Amicon Ultra-4 centrifugal filter device with a 50 kDa cutoff. To determine the degree of occupation, UV/Vis spectra of the proteins were measured in a quartz cuvette (Suprasil, Hellma, Müllheim, Germany) with a path length of 1 cm, using a Shimadzu (Duisburg, Germany) UV-2450 spectrometer [33].

### 3.4. In Silico Study Using YASARA

The homology models of SpH1 and SpH2 were built using YASARA structure ( www.yasara.org, 2020) [84]. the built models were further energy minimised before MD simulation and the docking study [85]. The structures of indole and 3-bromoindole were built with YASARA structure and energy-minimised using YASARA 2 force field before using in the docking experiments.

MD simulations were run for the homology models of both enzymes for 1 ns using YASARA structure [86]. The simulation cell was extended by 10 Å around the whole model and filled with randomly-oriented water molecules with a density of 0.99 g/mL. The force field AMBER14, with 10 Å for cut off and particle mesh Ewald (PME) for long-range electrostatic, was used for running the simulation. The stability of the homology models was monitored by the root mean square deviation (RMSD) of the alpha carbon of the models throughout the simulation time.

Docking of either indole or 3-bromoindole was performed by YASARA structure, using Autodock for the MD simulated models of SpH1 and SpH2 [87]. The simulation cell was defined around the whole homology model of both enzymes. Docking results were analysed based on the binding energy (B-factor) calculated by YASARA structure and the interaction of the substrate with the amino acid residues of the active site pocket.

The docked models with indole or 3-bromoindole were further subjected to MD simulation using the previously mentioned parameters.

The size of the active site pocket cavity was determined using the Computed Atlas of Surface Topography of proteins (CASTp) (http://sts.bioe.uic.edu/castp/, 2020) for the free and docked homology models before and after MD simulation [88].

## 4. Conclusions

Flavin-dependent halogenases are the focus of research because of their enormous potential for the regioselective halogenation of organic compounds, leading to products with profound biological activity. In the present study, using our hidden Markov model

(pHMM), we screened bacterial associates of the *B. braunii* consortia (PRJEB21978) and identified 25 complete and partial putative, flavin-dependent halogenase genes. Interestingly, all newly identified genes were found to form a distinct clade, together with previously characterised BrvH, potentially indicating a similar substrate preference. Two selected flavin-dependent halogenases (SpH1 and SpH2), derived from one gene cluster of *Sphingomonas* sp., were subjected to a screening with different compounds, and 26 substrates were found to become halogenated (Figure 4). In accordance with recent findings, as well as with the activity performance of BrvH, both enzymes were observed to prefer bromination over chlorination. Notably, SpH2 is capable of dibrominating substrates, while SpH1 can only monobrominate. Moreover, the simulation in silico using YASARA homology models revealed that SpH1 possesses a deeper but less flexible active site pocket, while SpH2 features a wider and more flexible active site pocket. This explains the ability of SpH2 to halogenate bulkier substrates, as well as catalyse dibromination. Both were identified in one gene, which suggests that the newly identified flavin-dependent halogenases could act in concert and interact with one another. Likewise, other known flavin-dependent halogenases originating from one gene cluster, such as the correlates PrnA/PrnC, Bmp2/Bmp5, or KtzR/KtzQ, act in concert for the biosynthesis of multiple halogenated compounds. SpH1 and SpH2 accepted, in total, 26 different indole and phenol derivatives. Interestingly, both accepted 4-*n*-hexylresorcinol (**18**), tryptophol (**20**), and 3-(3-indolyl)propionic acid (**21**). However, the natural substrates of SpH1 and SpH2 remain unknown so far. In the gene cluster of both enzymes, a catecholate siderophore receptor was detected, which might give a hint towards their substrate. This study shows that screening with the HMM may discover many novel flavin-dependent halogenases in natural metagenomes. The algorithm detects mostly flavin-dependent halogenases from a clade similar to BrvH. The phylogenetic tree demonstrates that the detected novel, putative, flavin-dependent halogenases cluster within the clade of BrvH.

**Supplementary Materials:** Further data is available online at https://www.mdpi.com/article/10.3390/catal11040485/s1. Amino acid sequences of the novel identified F-Hals SpH1 and SpH2. SDS-PAGE with heterologously expressed SpH1 and SpH2; RP-HPLC (280 nm) of the enzymatic halogenation of SpH1 (A) and SpH2 (B) of indole to bromoindole in the presence of catalase. Absorption bands of FAD bound in SpH1 and SpH2 during FAD reconstitution. HPLC trace of the enzyme reaction with SpH1_K81A and indole. Characterization of the brominated substrates by SpH1 and SpH2 via mass spectrometry. Determination of regioselective halogenation site via NMR analysis. List of detected secondary metabolite biosynthetic gene clusters within the MAG 21. Area and volume of the active site pocket for Apo SpH2 and SpH1, docked with indole and 3-bromoindole before and after MD simulation.

**Author Contributions:** Conceptualization, P.R.N., O.B.-K. and N.S.; data curation, P.R.N.; formal analysis, P.R.N.; funding acquisition, O.K. and N.S.; investigation, P.R.N., O.B.-K., L.P. and M.I.; project administration, O.K. and N.S.; resources, O.K. and N.S.; supervision, O.K. and N.S.; validation, P.R.N., O.B.-K., L.P. and M.I.; visualization, M.I.; writing—original draft, P.R.N.; writing—review and editing, P.R.N., O.B.-K., L.P., M.I. and N.S. All authors have read and agreed to the published version of the manuscript.

**Funding:** This research was partly funded by the Deutsche Forschungsgemeinschaft (SFB 1416/1-2020) and European Community's Seventh Programme for research, technological development, and demonstration under grant agreement no. FP7-311956 (relating to the project "SPLASH–Sustainable PoLymers from Algae Sugars and Hydrocarbons"). The authors acknowledge the financial support of the German Research Foundation (DFG) and the Open Access Publication Fund of Bielefeld University for the article processing charge.

**Acknowledgments:** The authors thank Karl-Heinz van Pée, TU Dresden, for donation of the plasmid pClBhis–prnF encoding the flavin reductase PrnF from *Pseudomonas fluorescens*, and Prof. Dr. Werner Hummel, Bielefeld University, for the plasmid vector pET-21 ADH encoding the alcohol dehydrogenase from *Rhodococcus ruber* (*RR*-ADH).

**Conflicts of Interest:** The authors declare no conflict of interest.

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
