# Peer review of "Two Novel, Flavin-Dependent Halogenases from the Bacterial Consortia of Botryococcus braunii Catalyze Mono- and Dibromination"

_catalysts, doi:10.3390/catal11040485_

Round 1

Reviewer 1 Report

In this study authors utilizing the pHMM model, identified two new flavin dependent halogenases named SpH1 and SpH2. Their investigation is of great interest nowadays and contributes to the selective halogenation of organic compounds, highlighting the importance of bioinformatic tools for the discovery of newly enzymes.

My comments for the authors are listed below.

Lines 140-143: These sentences seem that are not part of the manuscript. Authors should correct this.

In Figure 1. red correction underline should be removed from the Trs-Hals.

In Figure 2. many of the explanations of the abbreviations are with a red correction underline. Please correct.

The amino acid sequence of SpH1 and SpH2 is missing. Please add in the SI.

SDS-Gels with the expression level of the enzymes in E. Coli should be added in SI.

In line 313-314 please rephrase: A homology model built using Yasara structure was used for both enzymes as no crystal structures are available.

In line 329, authors claim expansion of the active site pocket of SpH2 after docking of indole, however, results in Figure 5 show that the size of the active site pocket has been decreased. Authors should rephrase this and explain the observed result.  

In Figure 7 authors should rephrase to: In both cases a H-bond is formed….

Author Response

Reviewer 1:

Lines 140-143: These sentences seem that are not part of the manuscript. Authors should correct this.

Answer: We completely agree with the reviewer (and apologise!) and removed the relevant part

In Figure 1. red correction underline should be removed from the Trs-Hals. In Figure 2. many of the explanations of the abbreviations are with a red correction underline. Please correct.

Answer: We are grateful to the reviewer for the accurate inspection and have corrected the figures. This is only due to the correction feature of Word.

The amino acid sequence of SpH1 and SpH2 is missing. Please add in the SI.

Answer: We added the amino acid sequences to the SI.

SDS-Gels with the expression level of the enzymes in E. Coli should be added in SI.

Answer: We have added the images of two SDS gels showing the heterologous expression of the proteins SpH1 and SpH2 to the SI.

In line 313-314 please rephrase: A homology model built using Yasara structure was used for both enzymes as no crystal structures are available.

Answer: Good point, we have modified this sentence accordingly.

In line 329, authors claim expansion of the active site pocket of SpH2 after docking of indole, however, results in Figure 5 show that the size of the active site pocket has been decreased. Authors should rephrase this and explain the observed result.  

Answer: Thank you for the thorough review! The sentence was incorrectly phrased and we have modified it accordingly in the revised manuscript. The claim was for the expansion of the pocket after MD simulation -as the docked indole would take space within the active site pocket- which clearly shows how flexible is the active site pocket of SpH2 compared to SpH1 in either cases of indole and 3-bromoindole. The explanation for the reduced area in SpH2 might be attributed to the difference in geometry of the active site pocket and cavity area.

In Figure 7 authors should rephrase to: In both cases a H-bond is formed….

Answer: We have rephrased this sentence.

Reviewer 2 Report

Comments to Manuscript:

1) Page 2, Line 69:     The literature reference 23 for Yeh et al. is incorrect.     

2) Page 2, Line 92:      It should be clarified whether something is missing or the “and” should be deleted in the formulation “... , while and Smith et al. ....”.

3) Page 3, Line 141:   The sentence “This section may be divided by subheadings” and the following sentence are not needed.     

4) Page 5, Line 160:   This formulation should be improved.            

5) Page 6, Line 198:   This sentence is incomplete.   

6) Page 9, Line 299:   ....    chlorocatechelins ....

7) Page 10, Line 347: Some more details should be given how the binding affinities were obtained. The described dimension kjol/mol for the binding affinities is not commonly used.       

8) Page 11, Line 386: .... linear gradient ...     

9) Page 12, Line 428:  Gene synthesis ....      

10) Page 13, Line 456:  The unit definitions for the enzyme activities should be given here.

11) Page 13, Line 459:  The SI prefix for millimolar concentrations should be used for the substrate concentration here as well as for the concentrations  throughout manuscript.        

12) Page 13, Line 464:   The type of catalase used should be specified.

Author Response

1) Page 2, Line 69:     The literature reference 23 for Yeh et al. is incorrect.

Answer: Thank you for the careful review! We corrected the reference.

2) Page 2, Line 92:      It should be clarified whether something is missing or the “and” should be deleted in the formulation “... , while and Smith et al. ....”.

Answer: Thanks for your close reading. We changed this part.

3) Page 3, Line 141:   The sentence “This section may be divided by subheadings” and the following sentence are not needed.

Answer: We completely agree with the reviewer and have removed the relevant part.

4) Page 5, Line 160:   This formulation should be improved.

Answer: We modified the relevant part as requested.

5) Page 6, Line 198:   This sentence is incomplete.

Answer: Thank you for the careful review! We changed this part.

6) Page 9, Line 299:   ....    chlorocatechelins ....

Answer: Thanks for your careful inspection. We changed this part.

7) Page 10, Line 347: Some more details should be given how the binding affinities were obtained. The described dimension kjol/mol for the binding affinities is not commonly used.

Answer: Thank you for the thorough review! We have now included the explanation for the determination of specific activity and have modified the material and method part accordingly.

8) Page 11, Line 386: .... linear gradient ...

Answer: Thank you for your attentive reading. We changed this part.

9) Page 12, Line 428:  Gene synthesis ....

Answer: We changed this part.

10) Page 13, Line 456:  The unit definitions for the enzyme activities should be given here.

Answer: We included the calculation formula in the Material and Methods part.

11) Page 13, Line 459:  The SI prefix for millimolar concentrations should be used for the substrate concentration here as well as for the concentrations throughout manuscript.

Answer: We checked the revised manuscript regarding the use of the SI prefix

12) Page 13, Line 464:   The type of catalase used should be specified.

Answer: Catalase from bovine liver lyophilized powder (2.000-5.000 units/mg protein; Sigma Aldrich, Germany) was used.

Reviewer 3 Report

The authors describe the search for flavin-dependent halogenases by association with Botryococcus brauni. The investigation is thorough and results are supported by data. This manuscript would be of interest to those seeking new ways of making halogenated organic compounds. This reviewer recommends reconsideration after major revisions.

Here are my comments to address:

  1. You differentiate the mechanism of flavin-dependent and halogenating peroxidases. This part is well-supported. On the other hand, it does not indicate the the enzyme is flavin-dependent. Did you run any experiments without the addition of flavin to confirm this statement?
  2. A figure needs to be added to show the mechanism of flavin-dependent halogenases and halogenating peroxidases. This would make the text more clear in your presentation.
  3. The kinetic measurements in Line 303 show activity of 4.2mU/mg for indole and 3.8mU/mg for bromoindole. What can be said about these values? I am not sure you can say for sure that there is higher specific activity with 4.2 vs. 3.8 without running multiple experiments. What is the standard deviation?
  4. Docking of bromoindole resulted in binding affinities for SpH1 and SpH2 that were very close (22.5 vs. 19.4). I am not sure that the claim that SpH2 is able to dihalogenate is indicated because of these in silico docking experiments.
  5. Line 543, there is a hanging sentence.

Author Response

  1. 1. You differentiate the mechanism of flavin-dependent and halogenating peroxidases. This part is well-supported. On the other hand, it does not indicate the enzyme is flavin-dependent. Did you run any experiments without the addition of flavin to confirm this statement?

Answer: Thank you very much for your comment. In heterologous expression in E. coli, the enzymes are already able to bind flavin. In experiments without additional flavin, we observed a lower halogenation activity. This was also observed for previously known flavin-dependent halogenases like RebH.

A figure needs to be added to show the mechanism of flavin-dependent halogenases and halogenating peroxidases. This would make the text more clear in your presentation.

Answer: We respectfully disagree. A figure displaying the mechanisms would not clarify the presentation. Instead, we added a short description:

The newly identified brominating enzymes belong to the class of flavin-dependent halogenases. Both enzymes have the characteristic FAD binding module GxGxxG, the conserved lysine residue (K85 and K81, resp.) and the WxWxIP motif suppressing monooxygenase activity. The reduced flavin (FADH2) reacts with molecular oxygen, forming FAD(C4a)-peroxide that in turn reacts with halide ions under HOX formation.

In contrast, heme- or vanadium-dependent haloperoxidases require hydrogen peroxide and release the halogenating species hypohalous acid HOX into the medium, which leads to unselective halogenation.

2. The kinetic measurements in Line 303 show activity of 4.2mU/mg for indole and 3.8mU/mg for bromoindole. What can be said about these values? I am not sure you can say for sure that there is higher specific activity with 4.2 vs. 3.8 without running multiple experiments. What is the standard deviation?

Answer: We are grateful for the comment, which pointed out that we did not explain the determination of the specific activity. All measurements were carried out in triplicate. We had deviations for SpH1 and the substrate indole ±  0.7 and for SpH2 and indole: ±. 0.03 and 3-Br-indole: ± 0.45. Additionally, we have modified the revised manuscript, because the differences are not significant.

3. Docking of bromoindole resulted in binding affinities for SpH1 and SpH2 that were very close (22.5 vs. 19.4). I am not sure that the claim that SpH2 is able to dihalogenate is indicated because of these in silico docking experiments.

Answer: Thank you for your comments to improve our manuscript. However, the in silico experiments did not claim the dihalogenation of indole based on the binding affinity; this was only one observation. In fact, the flexibility shown by SpH2 was the most intriguing part, where the pocket was expanded after MD simulation (with indole and 3-bromoindole) in case of SpH2, which is not the case with SpH1. The in silico observations were supported by the in vitro assays that demonstrated the di-bromination activity of SpH2.

4. Line 543, there is a hanging sentence.

Answer: We have changed this sentence accordingly.

Round 2

Reviewer 3 Report

The authors have significantly revised their manuscript according to reviewer comments and this reviewer recommends publication.

Author Response

Thank you very much for your recommendation.